# Differential processing of HIV envelope glycans on the virus and soluble recombinant trimer

Liwei Cao[1,2,3], Matthias Pauthner [2,3], Raiees Andrabi[2,3], Kimmo Rantalainen[3,4], Zachary Berndsen[3,4], Jolene K. Diedrich[1], Sergey Menis[2,3], Devin Sok[2,3], Raiza Bastidas[2,3], Sung-Kyu Robin Park[1], Claire M. Delahunty[1], Lin He[1], Javier Guenaga[2,3], Richard T. Wyatt[2,3], William R. Schief [2,3], Andrew B. Ward[3,4], John R. Yates III [1], Dennis R. Burton[2,3,5] & James C. Paulson [1,2,3]

As the sole target of broadly neutralizing antibodies (bnAbs) to HIV, the envelope glyco-protein (Env) trimer is the focus of vaccination strategies designed to elicit protective bnAbs in humans. Because HIV Env is densely glycosylated with 75–90 N-glycans per trimer, most bnAbs use or accommodate them in their binding epitope, making the glycosylation of recombinant Env a key aspect of HIV vaccine design. Upon analysis of three HIV strains, we here find that site-specific glycosylation of Env from infectious virus closely matches Envs from corresponding recombinant membrane-bound trimers. However, viral Envs differ sig-nificantly from recombinant soluble, cleaved (SOSIP) Env trimers, strongly impacting anti-genicity. These results provide a benchmark for virus Env glycosylation needed for the design of soluble Env trimers as part of an overall HIV vaccine strategy.

[1] Department of Molecular Medicine, The Scripps Research Institute, La Jolla, CA 92037, USA. [2] Department of Immunology and Microbiology, The Scripps Research Institute, La Jolla, CA 92037, USA. [3] Center for HIV/AIDS Vaccine Immunology and Immunogen Discovery and IAVI Neutralizing Antibody Center, The Scripps Research Institute, La Jolla, CA 92037, USA. [4] Department of Integrative Structural and Computational Biology, The Scripps Research Institute, La Jolla, CA 92037, USA. [5] Ragon Institute of MGH, MIT and Harvard, Cambridge, MA 0239, USA. These authors contributed equally: Liwei Cao, Matthias Pauthner  Correspondence and requests for materials should be addressed to J.C.P. (email: jpaulson@scripps.edu)

Although there is not yet an effective vaccine for the human immunodeficiency virus (HIV), broadly neutralizing antibodies (bnAbs) from chronically infected patients can protect against infection[1,2]. All bnAbs to date target the envelope glycoprotein (Env), which has become the primary target for design of a protective vaccine. A major barrier to HIV Env-based vaccine design is the glycan shield, comprising 26–30 N-linked glycans that cover the protein, thus blocking recognition by the immune system. Many bnAbs, have epitopes that are both protein and glycan dependent[3,4], while others have features that accommodate bulky glycans adjacent to their epitopes[5]. Thus, it is believed that Env-based immunogens with glycosylation matching authentic viral Env will be required at some stage in an overall vaccine strategy[6].

A major advance towards engineering an HIV Env-based vaccine was the development of stabilized soluble trimers[7–9]. These stable constructs contain the conformational and quaternary epitopes for many bnAbs that are not found on recombinant gp120 monomers, while shielding epitopes of many non-neutralizing antibodies that reside in the interface between monomers[3,7]. In general, the binding affinity of bnAbs to soluble trimers assessed in ELISA-based assays is predictive of neutralization potency to the corresponding virus, but there are exceptions for reasons that are not completely understood[7,10].

Several reports suggest that the reactivity of bnAbs can be dramatically affected by the structure of the N-glycans in their epitope[11,12]. The structural diversity in N-glycans arises from a biosynthetic pathway that starts with the transfer of a high mannose-type glycan ($Glc_3Man_9GlcNAc_2$) to Asn of each glycosite (Asn-X-Thr/Ser), followed by trimming of glucose and mannose residues to the common $Man_3GlcNAc_2Asn$ core and addition of terminal sugars to form complex-type glycans[13]. Analysis of soluble HIV Env trimers reveals that N-glycans have predominately high mannose-type glycans at some sites, and predominately complex glycans at other sites, reflecting minimal and extensive processing at the different glycosites, respectively[14–16].

Such differences are highly relevant to the specificity and antigenicity of bnAbs that include either high mannose or complex-type glycans into their epitopes[3,17]. For soluble well-formed trimers, complex glycans are enriched in the gp41 region, while patches of glycans on gp120 have mainly high mannose-type glycans, attributed to the dense cluster of glycans and steric constraints imposed by the quaternary structure[14,18]. A recent report on gp120 from Env derived from HIV grown in human lymphocytes assessed the types of glycoforms found at each site[16]. Although the abundance of each glycoform was not determined, 14 out of 24 glycosites contained mostly high mannose glycoforms, while others contained mainly complex-type or a mixture of complex, hybrid and high mannose-type glycoforms[16].

Given the importance of glycans on the specificity and antigenicity of bnAbs, we have compared the site-specific glycosylation of Env from three strains of infectious HIV (JR-FL, BG505, and B41) and recombinant membrane bound and soluble trimers from the same strains using a semi-quantitative method that determines the proportion of oligomannose, complex-type or no glycan at each glycosite[14,19]. For each strain, we found that site-specific glycan processing of Env from infectious virus produced in peripheral blood mononuclear cells (PBMCs) is similar to that of the recombinant membrane-bound trimers produced in HEK 293 F cells. In contrast, there were significant differences in processing of the N-glycans in the soluble Env trimers relative to PBMC-derived viral Env. Comparative analysis of a panel bnAbs for their ability to neutralize and bind each of the three viruses and the corresponding soluble SOSIP trimers, showed that the observed differences in glycosylation in fact strongly impact antigenicity, emphasizing the need to consider native Env glycosylation in HIV vaccine design.

## Results

**Isolation of mature cleaved HIV Env trimers**. HIV Env trimers from three HIV strains, JR-FL (subtype B), BG505 N332 (subtype A), and B41 (subtype B), were produced as recombinant soluble SOSIP.664 (SOSIP) trimers, membrane-bound ΔCt trimers, pseudovirus and/or infectious virus. The relationship between the various forms of Env is illustrated for JR-FL in Fig. 1a, and for B41 and BG505 in Supplementary Figures 1, 2, 3 and 4. All recombinant forms of Env and pseudovirus were produced in human embryonic kidney (HEK) 293F and 293T cells, respectively. Infectious virus was produced in 5–10 liters of cultured peripheral blood mononuclear cells (PBMCs). All HIV Env trimers were stabilized and purified using the bnAb PGT151[3,20], followed by size exclusive chromatography, to ensure isolation of well-formed functional, cleaved Env trimers (Fig. 1b, c, and Supplementary Figure 5). Analysis by SDS-PAGE and Blue Native PAGE (BN-PAGE) confirmed that only cleaved, trimeric Env was purified from 293F cell membranes or infectious virus (Fig. 1d, e, and Supplementary Figure 6). Using these methods, we were able to obtain ~25–50 µg of each Env preparation for the analysis of site-specific glycosylation.

**Soluble SOSIP Env differs from glycosylation of viral Env**. Env samples were analyzed using a proteomics based mass spectroscopy method that uses sequential treatment with endoglycosidase H (Endo H) and protein N-glycosidase F (PNGaseF) to introduce mass signatures into peptides that contain oligomannose (hybrid) structures (+203 daltons), complex type structures (+3), or no glycan (+0)[14,19]. The specificities of Endo H and PNGase F have been shown to high mannose (hybrid) glycans and complex type glycans, respectively, according to previous study[19]. Mass spectrometry (MS) detection for peptides with N + 0, N + 3, and N + 203 modifications is similar during electrospray ionization (ESI)-MS analysis, allowing to semi-quantitatively assess site-specific N-glycan processing of glycoproteins[14,19,21]. In addition, multiple proteases are used to generate numerous peptides for each glycosite, allowing detection and semi-quantitative analysis of the proportion of the three-glycosylation states at each site, further permitting the use of statistical analysis to assess site-specific differences between Env samples[14,19].

Each recombinant Env was digested and analyzed in duplicate from two biological batches, while native trimers from infectious virus and pseudovirus were analyzed in duplicate from same biological preparation due to limited material. As observed previously for analysis of soluble SOSIP trimers[14], the majority of the glycosites in each Env sample were fully (>95%) occupied by glycan, and only a few sites were partially (5–55%) unoccupied (Fig. 2). A striking observation with the JR-FL Env from infectious virus produced in PBMCs was that each site contained either predominately oligomannose (8 glycosites) or predominately complex-type glycans (18 glycosites), reflecting either minimal or extensive processing of glycans at each site, respectively (Fig. 2a, Supplementary Figure 7a, and Supplementary Table 1). Remarkably, the pseudovirus Env and the recombinant membrane-bound Env (JR-FL ΔCT, Fig. 2a) showed high similarity to the infectious viral Env, despite being produced in 293T, 293F and PBMC cells, respectively. Site-by-site comparison revealed no statistically significant differences in glycosylation except at site N241. In marked contrast, the soluble SOSIP Env produced in 293 F cells showed striking differences. Relative to viral Env, significant differences were observed at 13 of

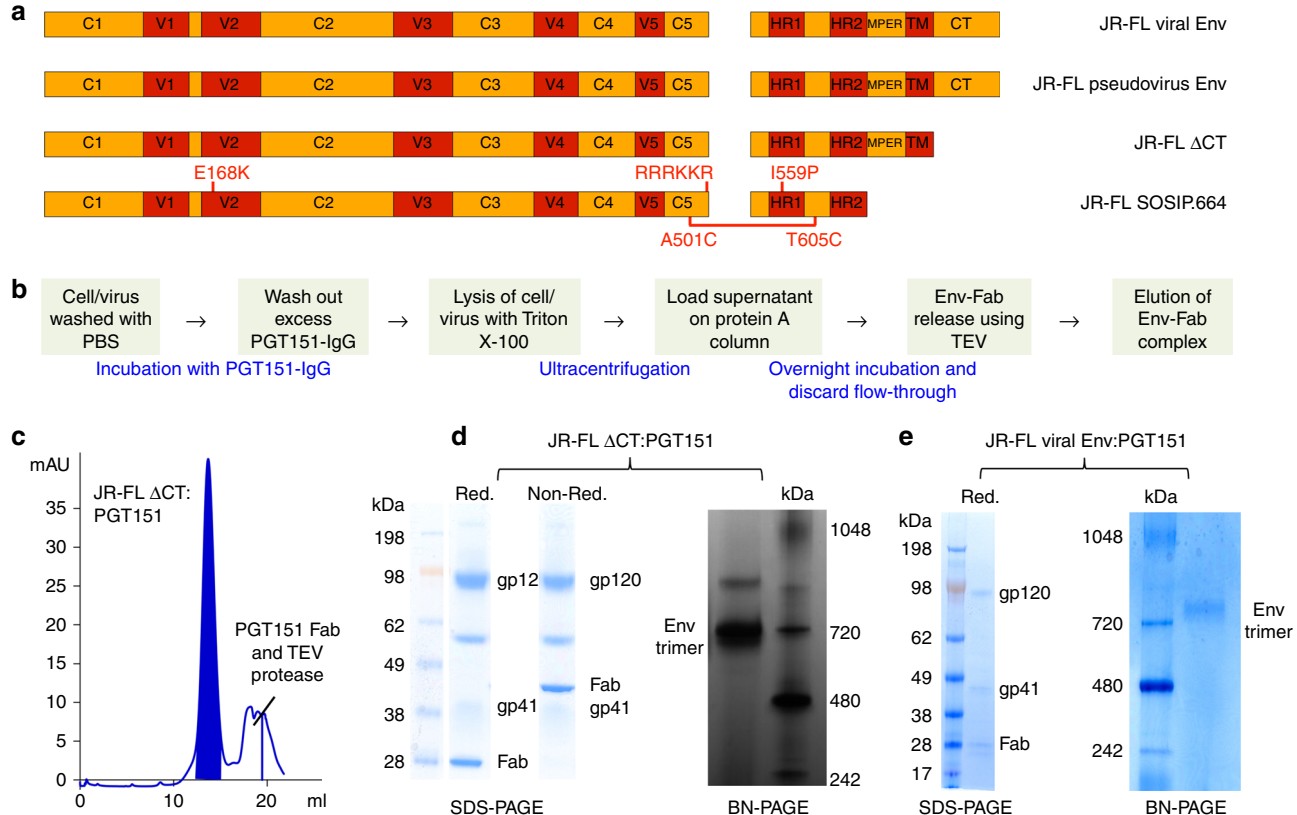

**Fig. 1** Design, purification, and characterization of JR-FL Env trimers. **a** Linear representation of JR-FL viral Env, JR-FL pseudovirus Env, JR-FL ΔCT, and JR-FL SOSIP.664. Modifications compared to the wild-type JR-FL gp160 sequence are indicated in red; **b** A schematic diagram of the method used for purification of well-ordered membrane-bound HIV Env trimer from the surface of 293F cells/HIV virus; **c** SEC chromatogram of the JR-FL ΔCT purification. The fraction that contained the JR-FL ΔCT-PGT151 complex was highlighted in blue; **d** SDS-PAGE (reducing and nonreducing) and BN-PAGE analysis of the JR-FL ΔCT trimer after purification; **e** SDS-PAGE (reducing) and BN-PAGE analysis of the purified JR-FL trimer derived from infectious HIV

the 26 glycosites (Fig. 2a). At these sites, the SOSIP Env typically had a mixture of oligomannose and complex glycans, while the viral Env displayed predominantly complex-type glycans (e.g., N88, N135, N160, N187, N241, N276, N301, N386, 637) or oligomannose-type glycans (N156, N339, N392), but no mixture of the two. In addition, several glycosites in the viral Env displayed higher glycan occupancy (N141, N637).

Similar analysis was performed for virus and recombinant Envs from the BG505 and B41 strains (Fig. 2b, c, Supplementary Figure 1, 3, 4, 7b, and 7c). For both strains, all glycosites could be robustly detected for the recombinant membrane-bound ΔCT constructs, which were used as references for comparisons with other Envs (Fig. 2b, c). Although wild-type BG505 virus has no glycosite at amino acid 332, the widely studied BG505 SOSIP has the N332 glycosite restored, to retain binding of V3-glycan directed bnAbs[7]. To make direct comparisons, we evaluated the membrane-bound ΔCT form with and without a glycosite present at amino acid 332. Processing at only a single glycosite, N411, appeared to be impacted by the presence or absence of glycans at Asn 332, which rendered N411 glycoforms predominantly oligomannose-type in N332 Env, but complex-type in T332 Env (Fig. 2b).

For both BG505 (Fig. 2b) and B41 (Fig. 2c), there were strong parallels with the glycosylation pattern of JR-FL derived Envs. For each strain, the viral and recombinant membrane-bound ΔCT Envs exhibited high occupancy at all glycosites, and had remarkably similar glycosylation profiles. In contrast, there were substantial glycosylation differences in the soluble SOSIP proteins relative to the corresponding recombinant membrane-bound

Env, with significant differences in the proportion of high mannose and complex-type glycans seen for 11 of 28 for BG505 and 14 of 29 glycosites for B41. For both strains, we observed a remarkable switch at site 301 from complex-type, in the viral and ΔCT Envs, to oligomannose, in the SOSIP Env. Other notable differences to the SOSIP-derived Envs were lower glycan occupancy and/or higher oligomannose content among the gp41 glycans (N611-N637), as well as more oligomannose glycans at the N88-N137, N187 and N197 glycosites. Of note, for BG505, the N276 glycan on the edge of the CD4-binding site[22] was partially processed on the viral and ΔCT Envs, but was comprised of exclusively under-processed oligomannose glycans on BG505 SOSIP Env.

Direct comparison of the glycosites for membrane-bound (ΔCT) and viral Envs for all three strains revealed conserved glycosites exhibiting predominately complex-type glycans (N88, N137/138, N185-187, N301, N355, N396-398, N462/463, N611, N616-618, N625, and N637), or oligomannose glycans (N156, N262, N295, and N362/363, Fig. 2, Supplementary Figure 8 and 9). Highly conserved glycosites with different processing included also N160 at the V2-apex, which was complex-type in JR-FL and oligomannose in the other two strains. Other conserved glycosites, including N276, N339, N386, and N392, likewise exhibited variations in glycosylation between strains.

To provide perspective on the site-specific glycan processing, we mapped all glycosites in JR-FL, BG505, and B41 Env trimers onto the structures of JR-FL ΔCT[23], BG505 SOSIP[24], and B41 SOSIP[25], respectively with color-coded glycans to distinguish sites that were predominantly high mannose (green), predominantly

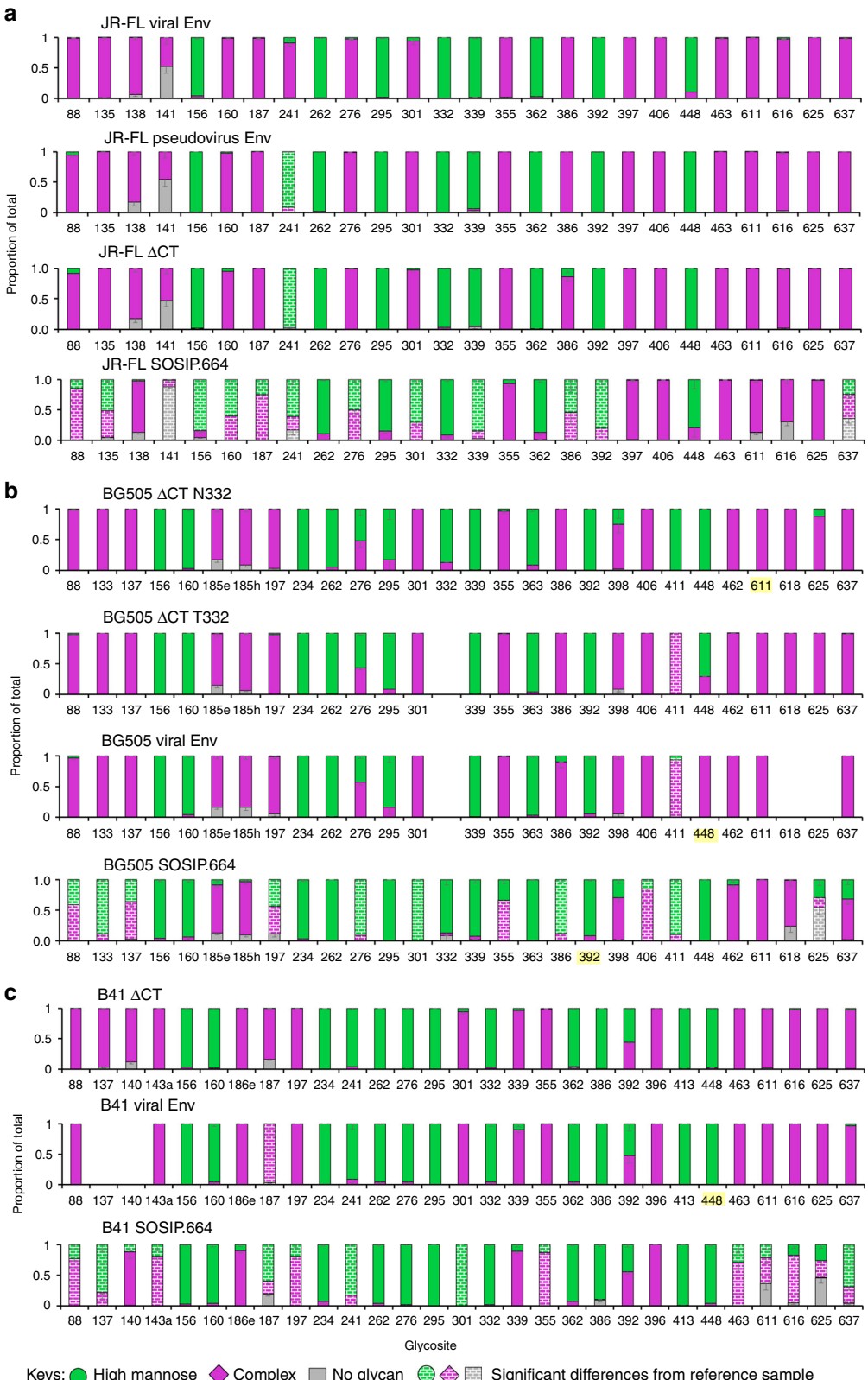

Keys: ● High mannose  ◆ Complex  ▪ No glycan  ● ◆ ▪ Significant differences from reference sample

complex-type (purple), or a mixed population of the two (*yellow*) (Fig. 3). This shows the remarkable similarity in the extent of glycan processing at each glycosite in virus and membrane-bound ΔCT versions of Env trimers, and significant differences in glycan processing at 11–14 of the 26–29 glycosites in the corresponding soluble SOSIP trimers. Relative to the viral Env proteins, soluble SOSIP Env from all three strains showed less glycan processing

and higher oligomannose content at the apex and at the gp120/gp41 interface.

**Membrane-bound and soluble Envs share a similar architecture.** To determine whether differences in structure between the soluble and membrane-bound forms of Env trimers could

**Fig. 2** Comparison of site-specific N-glycan processing of HIV-1 Env trimers. Site-specific analysis of glycan processing for **a** Env trimers from the JR-FL strain; **b** Env trimers from the BG505 strain; **c** Env trimers from the B41 strain. Shown for each glycosite were the proportion of peptides corresponding to unoccupied sites (grey), or sites containing high-mannose (green) and complex-type (purple) glycans. For glycosite numbers highlighted in yellow, proportions were assigned based on spectral hits since peak area did not reach the threshold. Sites with no number indicate the site was not present in the Env while sites with numbers and no bar graph indicate the site was not observed in LC/MS. Site-specific N-glycan processing of BG505 SOSIP is adapted with permission from ref. [14]. The glycosylation profiles of JR-FL viral Env, BG505 ΔCT N332 Env, and B41 ΔCT Env were used as a reference for comparison with other Envs from the corresponding strain. Significant differences from reference samples are shown as colored bricks. Each recombinant Env was digested and analyzed in duplicate from two biological batches, while native trimers from infectious virus and pseudovirus were analyzed in duplicate from the same biological preparation ($n = 6$). Differences were assessed for the proportions of no glycan, high mannose and complex type at each glycosite. Differences of >10% were determined to be significant if the $P$ value was <0.05 using a Mann–Whitney test. Mean ± s.e.m. were plotted

account for differences in glycan processing, we compared the cryo-EM reconstructions of both forms for all three strains. Accordingly, cryo-EM reconstructions of membrane-bound BG505 ΔCT T332 (subtype A) and B41 ΔCT (subtype B) in complex with PGT151 Fabs were determined at resolutions of 4.5 Å and 6.7 Å for comparison with the corresponding published soluble SOSIP trimers[25,26] (Fig. 4a, b, Supplementary Figure 10 and 11, and Table 1) and the soluble and ΔCT structures of JF-RL Env[23,27]. Consistent with the previous studies, both reconstructions bound PGT151 Fabs in an asymmetric manner with a stoichiometry of no more than two Fabs per trimer[3,23]. Because of this, the three gp140 interfaces were designated as interfaces A, B, and C (Supplementary Figure 12). Overall, the two reconstructions were highly similar, with PGT151 binding in an almost identical mode, despite the differences in subtypes (75% of sequence identity). Direct comparison of the EM reconstructions of BG505 ΔCT T332 (blue) and BG505 SOSIP (grey, PDB ID: 5ACO)[26] revealed minor differences, only at the trimer apex (Fig. 4c, d). The overall Cα RMSD between membrane-bound and soluble SOSIP trimers across different HIV strains were distributed around 2 Å (Supplementary Figure 12), indicating that membrane-bound trimeric Envs share a very similar structure to the corresponding soluble SOSIP trimers.

**Glycan network impacting site-specific glycan processing.** With the EM reconstructions of recombinant membrane-bound Env trimers for the three HIV strains in hand, we mapped site-specific glycan processing on the surface of corresponding cryo-EM density maps (Fig. 4e and Supplementary Figure 13). Comparison of the reconstructions with color-coded glycans suggests that the highly similar architectures of these trimers (Fig. 4a, b, Supplementary Figure 12 and 14) leads to similar processing of the glycans on the three strains except at glycosites N160, N276, N339, N386, and N392. At the trimer apex, the glycan N160 consisted predominantly of high mannose glycans on BG505 ΔCT T332 and B41 ΔCT, and in contrast, exclusively complex type structures were found at this glycosite on JR-FL ΔCT. In principle, fewer glycans at the apex of JR-FL ΔCT could result in less glycan crowding relative to BG505 ΔCT T332 or B41 ΔCT, which could in turn provide access to processing enzymes (Fig. 4e). In addition to the apex glycans N156 and topologically proximal N185h/187 glycan found in all three strains, B41 and BG505 contain an additional glycan (N185e/N186e) near the glycan N160, which could effectively shield it from the glycan processing machinery. Furthermore, lack of a glycan at position N197 is likely to induce more open conformations at the apex of JR-FL ΔCT relative to BG505 ΔCT T332, facilitating the processing of the N160 glycan (Supplementary Figure 14b, 14c and 14d). Significant differences in glycosylation were also found at glycosite N276 that resides adjacent to the CD4-binding site, ranging from predominantly or partially under-processed oligomannose in B41 and BG505 ΔCT to exclusively processed complex type structures in JR-FL ΔCT. Since JR-FL is the only strain

that lacks the N234 glycan site, we suggest that the corresponding glycan may restrict the glycan processing machinery at N276, resulting in under processed glycans in B41 and BG505. Another potential example of altered processing influenced by a neighboring glycan is the impact of the glycan at N406 on glycans of the high mannose patch, including N339, N386, and N392 (Fig. 4e and Supplementary Figure 13). Lack of the N406 glycan appears to facilitate the processing of the neighboring glycan N339 on B41 ΔCT, which is likely to impact accessibility of the glycan cluster formed by N386, N392, and N362/363 to the glycan processing machinery.

**Antigenicity of virus, membrane-bound and soluble trimers.** The dramatic differences seen in glycosylation of membrane-bound and soluble SOSIP trimers suggests the potential for differences in recognition by glycan-dependent bnAbs. To assess antigenicity, we evaluated a panel of 30 bnAbs for their ability to bind to JR-FL E168K, BG505 N332 and B41 soluble SOSIP trimers in an ELISA-binding assay, and their ability to neutralize the corresponding full-length Env-pseudoviruses in a TZM-bl neutralization assay (Fig. 5a). For JR-FL, neutralization assays were also performed with pseudovirus grown in the presence of kifunensine (+kif), which blocks glycan processing leaving oligomannose ($Man_9$) glycans at all glycosites[14]. The antibody panel covered all major antigenic regions present on SOSIP trimers, comprising the V3-loop base and surrounding glycans (V3-glycan), the apical V2 loop and N156/N160 glycans (V2-apex), the CD4-binding site, and the gp120/gp41 interface. For BG505, both the pseudovirus and the corresponding SOSIP trimer included the N332 glycan site to enable comparisons of V3-glycan bnAbs. Similarly, JR-FL pseudovirus and SOSIP proteins contained the E168K mutation, to allow for V2-apex bnAb comparisons.

We compared $EC_{50}$ binding-titers with $IC_{50}$ neutralization-titers as a rough estimate of the relationship between antigenicity on the soluble trimer and that on the virus, based on the assumption that neutralization potency is related to affinity for the virus-associated trimer[7,10]. Remarkably, the $EC_{50}/IC_{50}$ ratios ranged over 100,000-fold, with 2/3 having ratios >1, reflecting more potent neutralization, and 1/3 with ratios <1, reflecting less potent neutralization compared to binding (Fig. 5a). The most dramatic differences in all three strains were seen for V2-apex directed bnAbs (e.g. PG9, PG16, CH03, PGT145), which had high $EC_{50}/IC_{50}$ ratios. For JR-FL pseudovirus grown in the presence of kifunensine, which blocks mannose processing leaving $Man_9$-$GlcNAc_2$-Asn at all glycosites, PG9, PG16 and CH03 (but not PGT145) lost their ability to neutralize, demonstrating their dependence on specific V2-apex glycoforms, in keeping with earlier observations[28–30]. We further assessed glycan dependence of antibody binding to JR-FL ΔCT Env trimer expressed on the surface of 293 F cells grown in the presence and absence of kifunensine (Fig. 5b). Here, binding of PG9, PG16, PGDM1400 and CH03 to membrane-bound Env produced in the presence of kifunensine was dramatically reduced, while PGT145 binding was

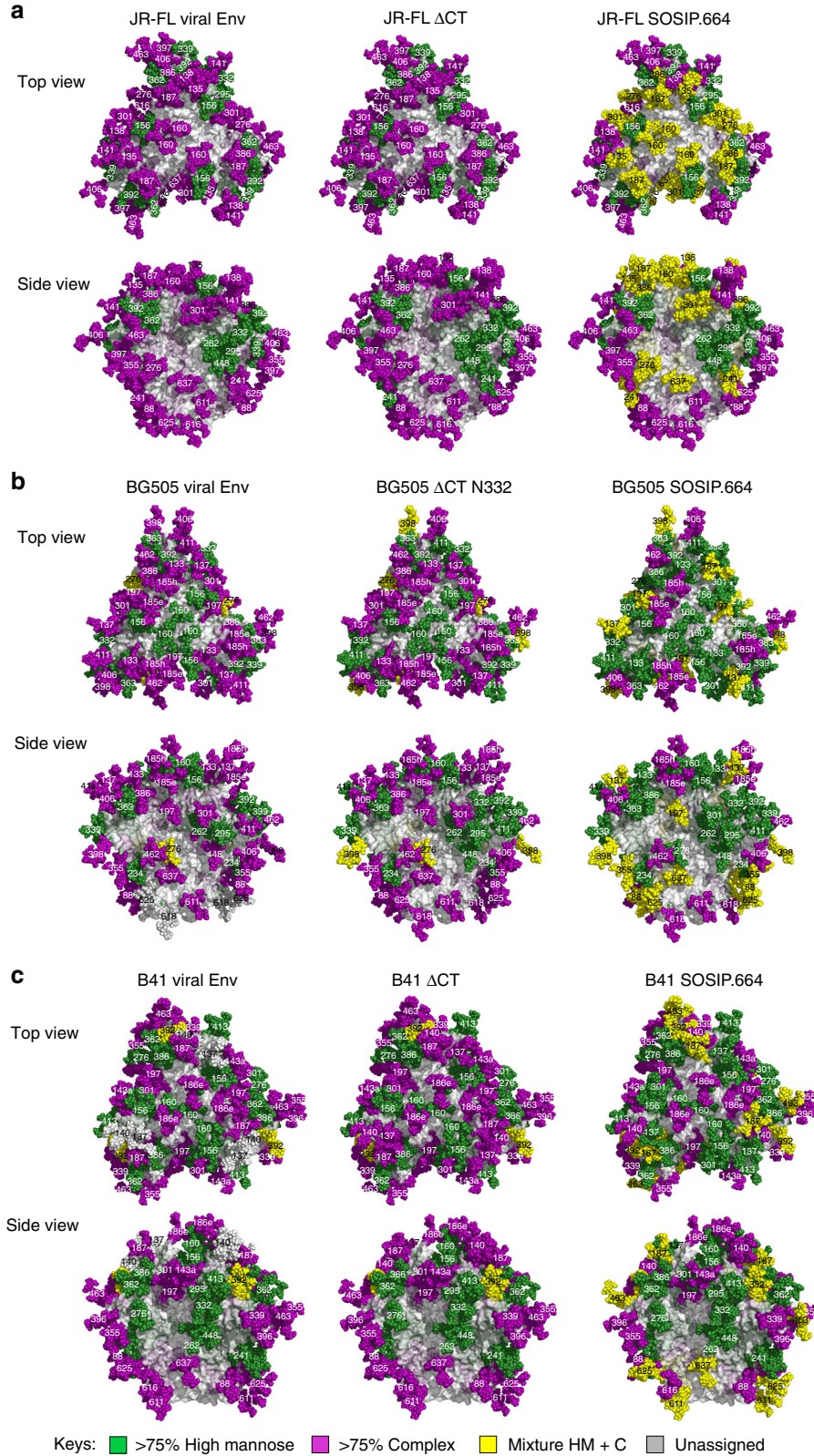

**Fig. 3** Mapping of site-specific N-glycan processing onto the structures of HIV-1 Env. **a** JR-FL strain; **b** BG505 strain; **c** B41 strain. The fully glycosylated models were constructed using JR-FL ΔCT (PDB: 5FUU), BG505 SOSIP (PDB: 5FYK), and B41 SOSIP. The surfaces of the trimers are shown in grey and the glycans are shown as spheres colored by proportion of oligomannose content at that site (Green spheres represent >75% high mannose glycosylation, purple spheres represent >75% complex type glycosylation, and yellow spheres represent the mixture of high mannose and complex type glycosylation (25%< high-mannose glycosylation <75%)).The glycans present at the glycosites that were not detected were shown as grey spheres

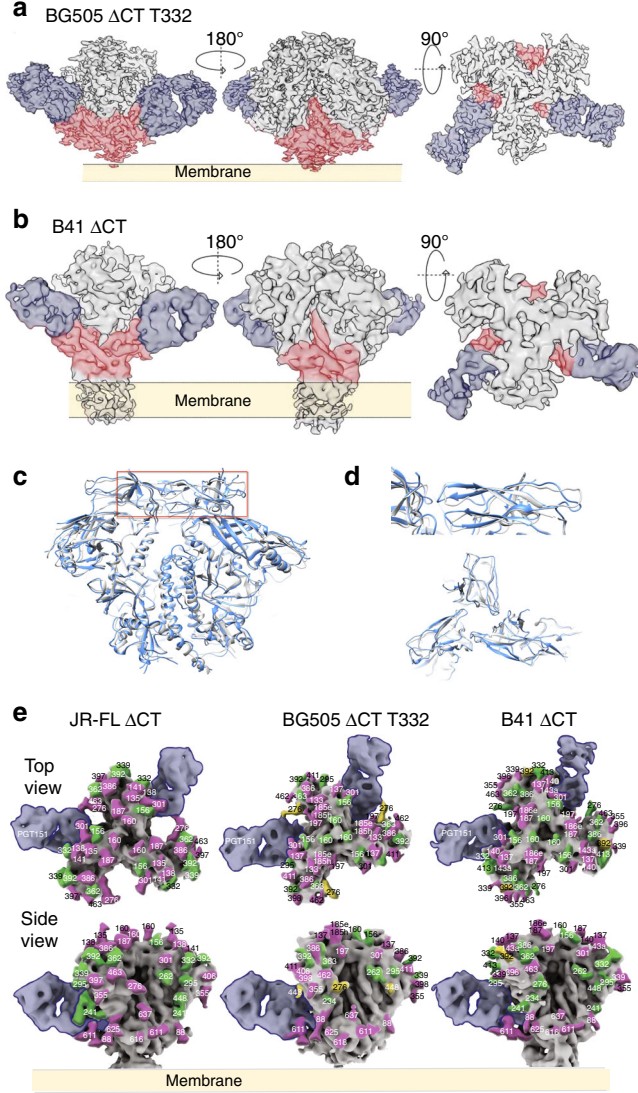

**Fig. 4** Comparison of Cryo-EM reconstructions of membrane-bound and corresponding soluble trimers. **a** Cryo-EM reconstruction of BG505 ΔCT T332 in complex with PGT151 Fab at 4.5 Å resolution. **b** Cryo-EM reconstruction of B41 ΔCT in complex with PGT151 Fab at 6.7 Å resolution. **c** Structural differences between BG505 ΔCT T332 and soluble BG505 SOSIP trimer (PDB: 5ACO). **d** Close-up of the V1/V2 region highlighted in **c**. **e** Mapping of site-specific glycan processing of JR-FL ΔCT, BG505 ΔCT T332, and B41 ΔCT onto the surface of the corresponding cryo-EM density maps with two PGT151 Fabs highlighted per trimer

slightly enhanced. Thus, for most of the V2-apex bnAbs, the increased processing of N-linked glycans at or near the apex in viral and membrane-bound Envs relative to the soluble SOSIP Envs corresponded to their large $EC_{50}/IC_{50}$ ratios.

Some antibodies to other antigenic regions also differed strongly between their binding to soluble SOSIP (ELISA) and neutralization of respective pseudovirus. For instance PGT151, which binds at the gp120/gp41 interface, exhibited high $EC_{50}/IC_{50}$ values for all three strains. Given the strong interactions of PGT151 with complex-type glycans at 611 and 637[23], it is notable that these glycans are significantly less processed in soluble SOSIP relative to the virus Env in all three strains. The complete loss of neutralization by PGT151 to JR-FL virus produced in the presence of kifunensine, and the loss of binding to JR-FL ΔCT Env produced in the presence of kifunensine (Fig. 5b)

underscores the dependence of this bnAb on complex-type glycans. For JR-FL the bnAbs that bind to the CD4 site do not seem to be glycan processing dependent, since there are only few differences when compared to neutralization of JR-FL produced in the presence of kifunensine. Finally, the V3-glycan bnAb, 2G12, bound soluble SOSIP and neutralized all strains equivalently, and furthermore neutralized JR-FL produced in the presence of kifunensine more potently than the wild-type virus. This is in keeping with oligomannose glycans as the primary epitope of 2G12[31,32].

## Discussion

Our comparison of site-specific glycosylation of the Env glycoprotein from infectious virus of three different HIV strains, JR-FL, BG505 and B41, revealed a highly distinct pattern of glycan processing, with each glycosite being either minimally processed with predominately oligomannose/hybrid glycans, or strongly processed with predominantly complex-type glycans. Impressively, the corresponding recombinant membrane-bound (ΔCT) Envs produced in 293F cells exhibited site-specific glycan processing strikingly similar to the viral Envs. However, it should be emphasized that while the degree glycan processing is similar between HEK 293F cells and PBMCs, these cell lines may produce complex glycans that differ in branching and terminal sialic acids. In contrast to the similarity of the virus and membrane-bound Envs, the soluble SOSIP trimers exhibited significantly different glycan processing at 11–14 of the 26–29 glycosites in each of the three strains. In most cases the differences reflect more complete processing of glycans to complex-type in the membrane-bound/viral Envs, including a complete switch from high-mannose-type to complex-type at N301 for all three strains.

The basis for the similarities in glycosite processing of recombinant (ΔCT) and viral membrane-bound Envs, and significant differences in processing of the glycosites in the corresponding SOSIP Envs remain to be determined. During their synthesis the soluble SOSIP trimer is released into the lumen of the ER while the membrane-bound (ΔCT) Env remains bound to the membrane. Thus, their topology are different during their transit through the ER and Golgi apparatus where the glycan-processing enzymes and glycosyltransferases are membrane-bound (Fig. 6). This difference could in principle impact the processing in a site-specific manner[33]. Other factors such as stabilizing mutations, the amount of glycoprotein expressed in each cell and the speed of transit through the secretory pathway could also be factors in differential processing[34].

The overall frequency of complex type glycans in viral/membrane-bound HIV Env is 54–68%. These levels are much higher than reported in previous analyses of HIV Env glycosylation (2–44%)[15,18,35–39]. Early glycomics analysis of HIV Env was done by releasing N-linked glycans with PNGase F and then evaluating the total glycan pool for the proportion of oligomannose/hybrid and complex type glycans to assess the degree of processing[34,36]. Initial analysis of Env from JRCSF HIV pseudovirus produced in PBMCs or HEK 293 T cells found that the glycans were predominately oligomannose with only trace amounts (2%) of complex glycans[34,36], concluding that native Env was predominately oligomannose. However, follow-up studies showed that altering the way the pseudovirus was produced increased the proportion of complex glycans, and that Env from representative clade A, B and C infectious virus strains produced in 293T cells or PBMCs showed much higher glycan processing, with 21–44% complex type glycans[34,35].

A key feature of our workflow is the use of PGT151 mAb for purification, which binds selectively to well-formed functional Env trimers[3,18]. This is particularly relevant to the isolation of

**Table 1 Cryo-EM data collection, refinement and validation statistics**

|  | BG505delCT in complex with PGT151 Fab (EMD-9062) (PDB ID 6MAR) | B41 in complex with PGT151 Fab (EMDB-9030) |
|---|---|---|
| Data collection and processing |  |  |
| Magnification | 41,666.7 | 48,543.7 |
| Voltage (kV) | 200 | 300 |
| Electron exposure (e–/Å$^2$) | 66 | 62 |
| Defocus range (μm) | −1.08:−5.3 | −0.67:−5.2 |
| Pixel size (Å) | 1.2 | 1.03 |
| Symmetry imposed | C1 | C1 |
| Initial particle images (no.) | 155,001 | 103,079 |
| Final particle images (no.) | 67,010 | 6574 |
| Map resolution (Å)  FSC threshold | 4.5 (0.143) | 6.7 (0.143) |
| Map resolution range (Å) | ≥4.5 | ≥6.7 |
| Refinement |  |  |
| Initial model used (PDB code) | 5FUU | N/A |
| Model resolution (Å)  FSC threshold | 4.2 (0.143) | N/A |
| Model resolution range (Å) | ≥4 | N/A |
| Map sharpening $B$ factor (Å$^2$) | −150 | −180 |
| Model composition  Non-hydrogen atoms  Protein residues  Ligands | 19218,2228,51 | N/A, N/A, N/A |
| $B$ factors (Å$^2$)  Protein  Ligand | N/A, N/A | N/A, N/A |
| R.m.s. deviations  Bond lengths (Å)  Bond angles (°) | 0.007,1.084 | N/A, N/A |
| Validation  MolProbity score  Clashscore  Poor rotamers (%) | 2.02,8.73,0.1 | N/A, N/A, N/A |
| Ramachandran plot  Favored (%)  Allowed (%)  Disallowed (%) | 89,11,0 |  |

Env from pseudovirus and infectious virus, since they are known to contain a mixture of functional cleaved Env trimers, and non-functional Env-froms, comprising both cleaved and uncleaved trimers as well as monomeric gp120[40–43]. Pritchard et al. conducted glycomics analysis on Env isolated from pseudovirus comparing purification with PGT151 and a panel of bnAbs that recognize both functional and nonfunctional forms of Env[20]. They found that Env purified with a panel of bnAbs (b12, b6, F425-b4e8, 2F5, and 4E10) contained a major fraction of gp160 that was absent from PGT151-purifed Env, and this fraction had a much higher content of oligomannose (>75%) than gp120 or gp41 preparations from the cleaved trimers. This suggests that the glycans on nonfunctional gp160 are much less processed (higher oligomannose). It is important to note that for well-formed Env trimers, PGT151 does not select specific glycoforms, as we and another group have previously shown for stable BG505 SOSIP preparations[14,18]. We found no differences in site-specific glycosylation of Env purified using HisTag/Ni$^{2+}$ column purification or three different carbohydrate-dependent antibodies (PGT151, PGT145 or 2G12)[14,18]. As a further test to determine the suitability for purification of virus Env, we found that, PGT151 completely neutralized virus from the JR-FL E168K and BG505 strains, and neutralized up to 75% of the B41 virus (Supplementary Figure 5), indicating it is very unlikely that PGT151 introduces bias for selection of glycoforms.

The proteomics based method we used provided the sensitivity to compare semi-quantitative analysis of site-specific glycan processing from infectious virus and recombinant forms of Env. Moreover, the method has been validated with regards to showing equivalent sensitivity for the three forms of a peptide reflecting no glycan, high mannose or complex type glycans[14,21]. Although glycoproteomics methods that analyze intact glycopeptides are inherently more qualitative, they can provide additional information about the glycoforms present at each site[15,16,35,37–39]. In this regard, it is notable that results from a very detailed glycoproteomics analysis of soluble BG505.664 SOSIP by Behrens et al.[15,35] were in high concordance with those obtained using our method (Fig. 2c)[14]. Go et al. used a similar glycoproteomics approach for analysis of soluble and membrane-anchored Env from a variety of strains[37,38], and Panico et al. performed a detailed glycomics and glycoproteomics analysis on the gp120 portion of Env from the replication-competent BaL HIV strain grown in a SUPT1-R5 T cell line[16]. A qualitative comparison of all these results revealed consensus complex-type glycans for 4 sites, high mannose-type glycans for 7 sites, and it was suggested that the other sites could be influenced by the genotype of the virus, producer cell type, or Env construct design[37].

Ultimately, it is desirable to understand how the structure of the functional cleaved Env influences site-specific processing of N-linked glycans. Based on the dramatic differences seen in

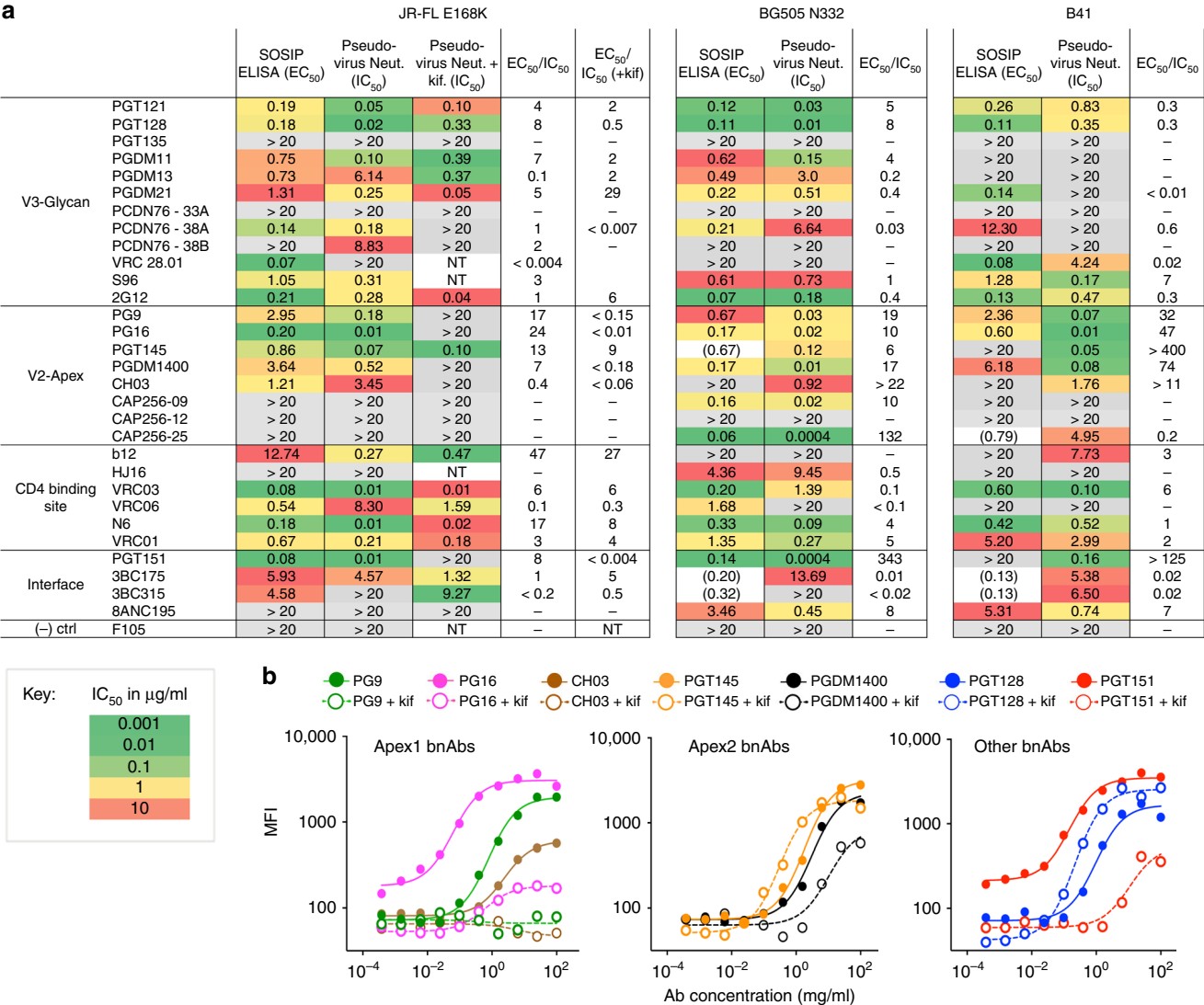

**Fig. 5** Comparison of the antigenicity properties of membrane-bound and soluble Env trimers. **a** Median IC$_{50}$ neutralization and EC$_{50}$ ELISA titers are provided in µg/ml and colored according to the listed scale for the three HIV-1 strains, including JR-FL E168K, BG505 N332, and B41. Neutralization of the full-length Env-pseudotyped viruses (IC$_{50}$) by bnAbs was determined in TZM-bl neutralization assays, while binding to the corresponding SOSIP trimers was measured by ELISA-binding assays. For JR-FL E168K, neutralization assays were also performed with pseudovirus grown in the presence of kifunensine (+kif), which blocks glycan processing leaving oligomannose (mainly Man$_9$) glycans at all glycosites. **b** Flow cytometric measurements for binding of fluorescent bnAb to membrane-bound JR-FL ΔCT Env on 293 F cells grown in the presence and absence of kifunensine. Green (pink, brown, orange, black, blue, or red) solid line represents PG9 (PG16, CH03, PGT145, PGDM1400, PGT128, or PGT151) binding to JR-FL ΔCT (-kifunensine), while green (pink, brown, orange, black, blue, or red) dashed line represents PG9 (PG16, CH03, PGT145, PGDM1400, PGT128, or PGT151) binding to JR-FL ΔCT (+kifunensine), respectively

glycosylation of soluble SOSIP.664 and membrane-bound/viral Env (Fig. 2), it is instructive to compare site-specific glycosylation of glycosites that are well conserved among Envs (Fig. 2, Supplementary Figure 8 and 9). On comparison of site-specific processing across the three HIV strains in this report, 30 glycosites were found in at least 2 of the strains (Fig. 2, Supplementary Figure 8), and of these 16 were complex-type and 6 were oligomannose type in each strain. There were 8 sites that exhibited different processing in the ΔCT Envs (N160, N241, N276, N339, N386, N392, and N411-413, 448), and only two sites differed in glycan processing between the ΔCT Env and the corresponding virus Env (N241 and N448). The degree to which site-specific, all or none glycan processing is consistent across these three strains is remarkable, and is very likely a result of common structural features of functional cleaved Envs, as well as similar exposure to

glycosidases and comparable retention times in the ER/Golgi for membrane-bound as compared to soluble proteins, respectively.

Most relevant to current HIV vaccine-design strategies is the impact of glycosylation on the antigenicity of Env and the desirability to induce glycan-dependent bnAbs[44]. As shown in this report, site-specific glycan processing and bnAb recognition of recombinant soluble Env differed significantly from the corresponding membrane-bound (ΔCT) and authentic viral Env for all three strains. Although not evaluated here, since the virus and recombinant Env were produced in PBMC and HEK293 cells, they may still produce complex glycans that differ in branching and terminal glycosylation[12]. For production of vaccines, although both HEK293 and CHO cells have been approved by the FDA, Chinese hamster ovary (CHO) cells are preferred for manufacturing biotherapeutics[45,46]. In this regard, is notable that

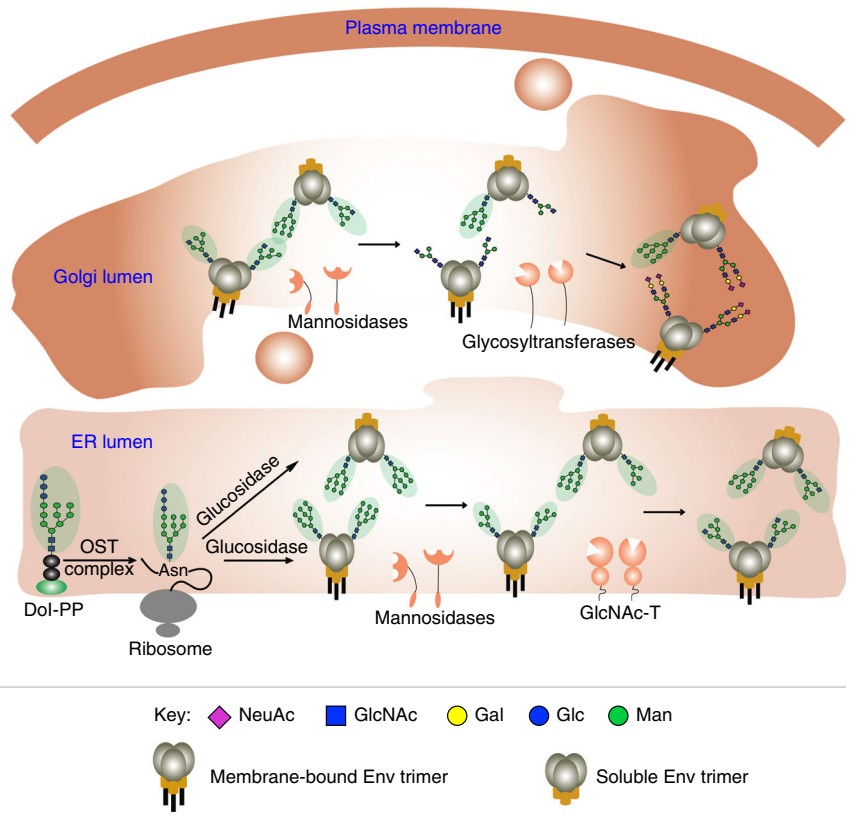

**Fig. 6** Topology of N-glycan processing for soluble and membrane-bound Env. Since the glycan processing enzymes are membrane bound in the ER and Golgi, the membrane bound and soluble Envs exhibit a different topology relative to the active sites of the enzymes. Glycan structures that are Endo H-sensitive are highlighted in green. Purple diamonds represent sialic acid, blue squares represent N-acetylglucosamine, yellow circles represent galactose, blue circles represent glucose, and green circles represent mannose

comparable glycan processing was observed for soluble BG505.664 SOSIP produced in HEK293 and CHO cells[47]. Ultimately, establishing the immunogenicity of vaccine constructs with glycosylation resembling authentic viral Env in animal models will help inform the overall strategy for development of a HIV Env-based vaccine.

## Methods

**Ethical approval.** The study was approved by The Scripps Research Institute Institutional Review Board with adherence to the appropriate informed-consent process.

**Expression of soluble SOSIP.664 (SOSIP) trimers.** JR-FL E168K, BG505 N332, and B41 SOSIP expression plasmids were transiently co-transfected with furin plasmids in a 2:1 ratio into 293 F cells (Invitrogen), using 293Fectin (Invitrogen), cultured in 293 Freestyle media (Life Technologies) and grown for 144 h at 37 °C. Culture supernatants were collected and sterile filtered using a 0.22 μm pore filter (Nalgene). Supernatants were purified using the following protocols.

**Purification of SOSIP trimers.** All SOSIP trimers used for glycosylation analysis and ELISA-binding experiments were purified as follows unless otherwise noted. N-terminally His-tagged JR-FL E168K, N-terminally Avi-tagged BG505 N332, and N-terminally Avi-tagged B41 SOSIP.664 constructs were transfected as described above. Supernatants were subsequently purified over PGT151 antibody affinity columns (for glycosylation analysis) or 2G12 antibody affinity columns (for ELISA experiments)[7]. Supernatants were slowly run over a CNBr-activated Sepharose 4 Fast Flow (GE) conjugated PGT151 or 2G12 antibody affinity column at 4 °C overnight, and washed with 1× PBS, followed by wash buffer (20 mM Tris-HCl (pH 8), 0.5 M NaCl). The Env trimers were eluted in 3 M Magnesium chloride and buffer exchanged into TBS. The eluate was then run over a Superdex 200 Increase SEC column (GE Healthcare) in TBS and trimer-size fractions were collected. The trimeric state of all proteins was confirmed by blue-native PAGE. Avi-tagged SOSIP trimers for ELISA-binding experiments were in vitro biotinylated using the BirA enzyme according to the manufacturer protocol (Avidity), in-between 2G12-antibody and size exclusion chromatography purification steps.

JR-FL E168K SOSIP was purified for ELISA-binding experiments by negative selection using F105 antibodies[48]. Culture supernatants were run over a Galanthus nivalis lectin (GNL) affinity chromatography column overnight at 4 °C and eluted with elution buffer (0.5 M Methyl a-D-mannopyranoside (Sigma), 0.5 M NaCl in PBS). The eluate containing SOSIP trimers was run over a Superdex200 10/300 GL SEC column to separate trimer-size oligomers from aggregates and gp140 monomers. Trimer-size JR-FL E168K SOSIP containing fractions were loaded onto a protein A agarose column previously loaded with 2-fold weight excess of non-neutralizing mAb F105 over the amount of trimeric SOSIP loaded. The column was gently rocked at 4 °C for 45 min, the solid phase was allowed to settle for 5 min and the well-ordered trimeric JR-FL SOSIP protein was recovered from the flow-through in a negative selection manner. The trimeric state of all proteins was confirmed by blue-native PAGE.

**Membrane-bound Env trimer expression.** JR-FL ΔCT, BG505 ΔCT N332, BG505 ΔCT T332, and B41 ΔCT expression plasmids were transiently co-transfected with furin plasmids in a 2:1 ratio[49] into 293F cells (Invitrogen), using 293Fectin (Invitrogen), cultured in 293 Freestyle media (Life Technologies) and grown for 72 h at 37 °C. Membrane-bound Env trimers were then isolated using PGT151-TEV pull down.

**Pseudovirus production.** Replication incompetent HIV pseudovirus was produced by co-transfecting env plasmids with an env-deficient pSG3Δenv (AIDS Reagents Program) backbone plasmids in HEK293T cells (ATTC) in a 1:2 ratio, using the X-tremeGENE 9 transfection reagent (Roche). Pseudoviruses produced in the presence of glycosylation inhibitors were generated by treating 293T cells with 25 μM kifunensine (Cayman Chemical Co.) on the day of transfection[50]. Pseudovirus was harvested after 72 h for by sterile-filtration of cell culture supernatants through a 0.22 μm membrane (Nalgene). Supernatant was then either frozen at −80 °C for neutralization assays or membrane-bound virion-derived Env trimers were isolated using PGT151-TEV pull down.

**Live virus preparation.** To grow replication-competent infectious viruses, full-length sequences encoding a clade A (BG505) and two clade B (JR-FL, B41) HIV-1 isolates were obtained from the National Institutes of Health (NIH), AIDS Research and Reference Reagent Program. Primary PBMC samples were obtained

through the commercial vendor AllCells (Emeryville, CA). To prepare virus stocks for infecting the peripheral blood mononuclear cell (PBMC), full-length virus-encoding plasmids were transfected into the 293T cells. The viral supernatants were collected 48 h post-transfection and the virus 50% tissue culture infectious doses (TCID50) were determined by titration in TZM-bl target cells. To grow PBMC-derived native viruses, PBMC were isolated by Ficoll-Hypaque gradient centrifugation from leukapheresis sample of a healthy HIV seronegative donor. The leukapak sample was obtained from a commercial vendor (AllCells) using a protocol approved by the Institutional Review Boards of both AllCells and The Scripps Research Institute. Cytotoxic CD8+ T-cells were depleted from PBMCs using the RosetteSep human CD8+ lymphocyte depletion kit (STEMCELL) following the manufacturer's instructions. Briefly, the depletion cocktail was added to PBMCs and incubated at room temperature for 20–30 min prior to layering and separation by Ficoll-Hypaque. The PBMC-viruses were grown in phytohemagglutinin (PHA)-activated cells[51]. PBMCs were activated by 3-day incubation in PHA (5 µg ml$^{-1}$ (Difco Laboratories)) and intereukin-2 (IL-2: 10 U ml$^{-1}$ (Roche)) supplemented RPMI 1640 medium. After the PHA-activation, the cells were washed and further cultured at 1 million cells/ml density in RPMI medium with IL-2 (20 U ml$^{-1}$). The 293T cell grown virus stocks were added at TCID50 virus dose to infect the PBMCs in T225cm flasks and the cell cultures were maintained in 7% $CO_2$ incubators at 37 °C. Viral supernatants were collected every 2-day for approximately 3-week and exchanged with IL-2 culture medium. The viral supernatants were made cell-free by centrifugation at 1000 × $g$ and filtration though 0.45-m tissue culture filters. The growth of the replicating viruses in the culture supernatants was monitored by p24-Ag levels.

**PGT151-TEV antibody expression and purification.** PGT151-TEV heavy chain and light chain expression plasmids[23] were transiently co-transfected in 1:1 ratios into 293 F cells (Invitrogen), using 293Fectin (Invitrogen), cultured in 293 Freestyle media (Life Technologies) and grown for 96 h at 37 °C. Culture supernatants were collected and sterile filtered using a 0.22 µm pore filter (Nalgene). Antibody supernatants were purified over Protein A Sepharose 4 Fast Flow (GE healthcare) columns[52].

**PGT151-TEV pull down purification of membrane-bound Env.** Recombinant membrane-bound Env trimers were purified with a modified PGT151-based method[3,23], followed by size exclusive chromatography (SEC). In summary, 2 liters of 293F cells expressing Env were washed once with cold PBS pH 7.4 and spun down at 3000 × $g$ for 15 min at 4 °C. The washed cells were resuspended in 30 ml of cold PBS, followed by adding 1 mg of PGT151 with TEV cleavage site. The resulting solution was incubated at 4 °C for 2 h in continuous rotation. The cells were spun down at 3000 × $g$ for 15 min at 4 °C, and excess PGT151 was washed out with cold PBS. The cells were then lysed with solubilization buffer (0.5% (w/v) Triton X-100, 300 mM NaCl, 50 mM Tris-HCl (pH 7.4)), and cell debris was spun down at 39,000 × $g$ for 1.5 h at 4 °C. The supernatant was incubated with Ultra Protein A resin (GenScript) overnight at 4 °C. After incubation, the solution was transferred to a gravity-flow column. The resin was washed with 100 ml of wash buffer 1 (50 mM Tris-HCl (pH 7.4), 300 mM NaCl, 0.1% (w/v) CHAPS, 0.03 mg ml$^{-1}$ deoxycholate), 100 ml of wash buffer 2 (50 mM Tris-HCl (pH 7.4), 500 mM NaCl, 0.1% (w/v) n-Dodecyl β-D-maltoside (DDM), 0.03 mg ml$^{-1}$ deoxycholate), 100 ml of gel filtration buffer (50 mM Tris-HCl (pH 7.4), 150 mM NaCl, 0.1% (w/v) DDM, 0.03 mg ml$^{-1}$ deoxycholate), and 40 ml of cleavage buffer (50 mM Tris-HCl (pH 7.4), 150 mM NaCl, 0.1% (w/v) DDM, 0.03 mg ml$^{-1}$ deoxycholate, 2 mM EDTA). The Env-PGT151 complex was released with 500 µg of TEV protease in 10 ml of cleavage buffer for 4 h at room temperature. The flow-through was collected and remaining Env was eluted with 40 ml of gel filtration buffer. Flow through and elute were pooled and concentrated to <500 µl using a 100 kDa MWCO concentrator (Millipore). The solution was applied to a Superose 6 column (GE Healthcare) that was pre-equilibrated with gel filtration buffer, and the fraction corresponding Env trimer was collected for the following studies.

Membrane-bound Env trimers derived from pseudovirus and infectious virus were purified by using the method described above except that pseudovirus and virus were spun down at 22,000 × $g$ for 1 h at 4 °C and particles were resuspended in <5 ml of cold PBS before addition of 1 mg of PGT151 with TEV cleavage site.

**Proteolytic digestion.** Approximately 25–50 µg of HIV-1 Env trimers were denatured with 8 M urea in 100 mM ammonium acetate (pH 6). The resulting proteins were reduced and alkylated with 10 mM of dithiothreitol (DTT, 56 °C, 1 h) and 50 mM of iodoacetamide (45 min, in the dark), respectively. The proteins were then divided into five aliquots and digested with triple digestion, chymotrypsin, and combination of trypsin and chymotrypsin[14,19].

Three out of five aliquots were digested following a triple digestion protocol. The first fraction was digested with Arg-C (Promega) at an enzyme/substrate ratio of 1:20 (w/w) in 100 µl of 100 mM ammonium bicarbonate containing 5 mM DTT and 0.2 mM EDTA (pH 8, 37 °C, 4 h), followed by trypsin (Promega) digestion at an enzyme-substrate ratio of 1:10 (w/w) in 100 mM ammonium acetate (pH 6, 37 °C, 16 h). The second fraction was digested with elastase (Promega) at an enzyme/substrate ratio of 1:20 (w/w) in 100 mM ammonium bicarbonate (pH 8, 37 °C, 16 h). The third fraction was digested with subtilisin

(Sigma) at an enzyme/substrate ratio of 1:20 (w/w) in 100 mM ammonium bicarbonate (pH 8, 37 °C, 4 h). The peptide mixtures derived from triple digestion were combined into a triple digestion sample.

The fourth fraction was digested with chymotrypsin (Promega) at an enzyme/substrate ratio of 1:13 in 100 mM ammonium bicarbonate (pH 8, 30 °C, 10 h).

The fifth fraction was digested with combination of trypsin and chymotrypsin at enzyme/substrate ratios of 1:20 (w/w) and 1:13 (w/w), respectively, in 100 mM ammonium bicarbonate (pH 8, 37 °C, 16 h).

The protease enzymes were denatured at 100 °C for 5 min before deglycosylation.

**Deglycosylation.** The resulting (glyco)peptides were then deglycosylated with the sequential treatment of Endo H and PNGase F to introduce mass signatures for peptide glycosites that were unoccupied or occupied by high mannose (hybrid) or complex type glycans.

The resulting (glyco)peptides were first deglycosylated with Endo H (New England Biolabs) at an enzyme/substrate ratio of 250 units per 10 µg in 20 µl of 100 mM ammonium acetate (pH 5.5, 37 °C, 1 h).

The (glyco)peptides were subsequently deglycosylated with PNGase F (New England Biolabs) at an enzyme/substrate ratio of 500 units per 10 µg in 100 mM ammonium bicarbonate (pH 8) prepared with O$^{18}$-water (97%, Sigma, 37 °C, 1 h).

**Mass spectrometric analysis.** All samples from HIV-1 Env proteins were analyzed on a Fusion Orbitrap tribrid mass spectrometer (Thermo Fisher Scientific). Approximately 1 µg of peptides were injected directly onto 75 µm i.d. capillary packed with 30 cm 1.7 µm BEH $C_{18}$ resin (Waters). The column was attached to a nLC 1000 (Thermo Fisher Scientific) and placed in line with the heated capillary of the Fusion Orbitrap tribrid mass spectrometer. The buffers used were A: 0.1% formic acid in 5% acetonitrile, and B: 0.1% formic acid in 80% acetonitrile. Peptides were separated using a 240-minute gradient as follows: 0–180 min, 1–25% B; 180–210 min, 25–40% B; 210–230 min, 40–90% B and hold at 90% B for a final 10 min of run time. The column was reequilibrated with buffer A prior to the injection of sample. As peptides were eluted from the microcapillary column, they were electrosprayed by the application of a distal 2.8 kV spray voltage. Peptides were detected by the mass spectrometer that was operated in a data-dependent mode. Full MS1 scans were collected in the Orbitrap at 120 K resolution with a mass range of 400–1500 m/z and an AGC target of 4e5. The cycle time was set to 3 s, and within this 3 s the most abundant ions per scan were selected for CID MS/MS in the orbitrap at 15 K resolution and with an AGC target of 1e6. Dynamic exclusion was enabled with exclusion duration of 5 s.

**Mass spectrometric data processing.** MS1 and MS/MS data were extracted from LC-MS/MS raw files by using RawConverter[53]. The MS/MS spectra were searched against the European Bioinformatic Institute (IPI) Bos taurus protein database using the ProLuCID algorithm (version 5, one component of Integrated Proteomics Pipeline-IP2)[54]. Of note, a composite database was used in the present study, which included original and reversed Bos taurus protein sequences as well as the sequences of HIV-1 Env trimers analyzed in this study. The reversed database was used to estimate peptide probabilities and FDRs. The parameters were set as: MS1 tolerance ≤50 ppm, MS2 tolerance ≤20 ppm, no enzyme specificity, carboxyamidomethylation (+57.02146 C) as a fixed modification, and oxidation (+15.9994 M), deamidation (+2.988261 N), GlcNAc (+203.079373 N), and pyroglutamate formation from N-terminal glutamine residue (−17.026549 Q) as variable modifications. The results were filtered by DTASelect (version 2.0)[55]. The parameters were set as: MS1 tolerance ≤10 ppm, minimum number of peptide per protein ≥2, and spectrum false positive rate ≤0.05. Since N-glycosylation occurs at a consensus motif (N-X-S/T, X can be any amino acid residue except proline), the remaining peptide sequences were further filtered to remove those peptides with N + 3 and/or N + 203 modifications that were not located at the motif. Census[56] (another component of IP2) was used to reconstruct chromatograms for identified peptides and calculate the abundance of peptide using peak area. Unidentified peptides were retrieved by retention time, isotope matching scoring, and accurate mass checking.

**Statistical analysis.** The following statistical analyses were performed with GraphPad Prism 5. The ion intensity peak area of the peptides identified from each raw file was generated by summing the peak area of this peptide over all charge states identified[14,15]. For each Env preparation, peak area data was summed for duplicate samples of all three protease conditions ($n = 6$). The summed values were used for statistical analysis with GraphPad Prism 5. A set of peptides with N + 0, N + 3, and N + 203 modifications were chosen only when at least one of the three was identified with peak area more than 1e7. Proportions of the three glycosylation states at a glycosite were generated based on spectra hits if no peptide for this glycosite met the threshold of 1e7. JR-FL viral Env, BG505 ΔCT N332, and B41 ΔCT were selected as references, and the site-specific glycan processing details of other HIV-1 Env trimers were compared to the corresponding reference by using a Mann–Whitney test. Significant differences were concluded only when the $P$ value was <0.05 as well as the difference on proportion of no glycan, high-mannose, or complex type glycans was >10%.

**SOSIP ELISA-binding assays**. Microlon 96-well plates (Corning) were coated overnight with either 2.5 µg ml$^{-1}$ streptavidin (Jackson ImmunoResearch) or 2.5 µg ml$^{-1}$ mouse anti-His antibody (Thermo Scientific) in phosphate-buffered saline (PBS) at 50 µl per well. Plates were then washed 4 times with PBS-Tween (0.05%) and blocked with PBS + 3% BSA for 1 h at room temperature. Avi-biotinlylated His-tagged JR-FL E168K SOSIP, Avi-biotinlylated BG505 N332 SOSIP, or Avi-biotinlylated B41 SOSIP protein was added at 1 µg ml$^{-1}$ in PBS + 1% BSA for 2 h at room temperature. Plates were then washed 4 times with PBS-tween (0.05%) and serially diluted monoclonal antibodies in PBS + 1% BSA were added for 1 h at room temperature (20 µg mL$^{-1}$ in 4-fold titrations). Plates were then washed 4 times with PBS-tween (0.05%) and alkaline phosphatase-conjugated goat anti-human IgG-Fc (Jackson ImmunoResearch, #109-055-098), was added for 1 h at a 1:1000 dilution (final concentration 0.33 µg ml$^{-1}$) in PBS + 1% BSA at room temperature. Plates were then washed 4 times with PBS-tween (0.05%) and 1 time with purified H$_2$O and absorption at 405 nm was measured following addition of phosphatase substrate in alkaline phosphatase buffer. EC$_{50}$ binding titers were calculated using Graphpad Prism v7.0. All EC$_{50}$ titers for antigenicity comparisons were measured in two or more independent experiments and were subsequently averaged.

**Pseudovirus neutralization assays**. Neutralization was tested by incubating wild-type pseudovirus, or pseudovirus produced in the presence of kifunensine, with serial dilutions of monoclonal antibodies for 1 h at 37 °C before transferring them onto TZM-bl cells (AIDS Reagent Program). Neutralization was measured in duplicate wells within each experiment. All IC$_{50}$ titers for antigenicity comparisons were measured in two or more independent experiments and were subsequently averaged. Neutralization IC$_{50}$ titers were calculated using the One site—Fit logIC$_{50}$ regression in Graphpad Prism v7.0. IC$_{50}$ titers of incomplete neutralization curves that reached at least 50%, but less than 90% maximal neutralization, were calculated by constraining the regression fit through 0 and 100% neutralization, to ensure accurate calculation of 50% neutralization titers.

**Flow cytometry binding assays**. Titrating amounts of mAbs (100 µg mL$^{-1}$ in 4-fold titrations) were added to HIV-1 env-transfected 293T cells (as described above) 2 days post-transfection and incubated for 1 h at 4 °C in 1× PBS[52]. After washing, cells were fixed with 2% para-formaldehyde (PolySciences) for 20 min at room temperature. Cells were then washed and stained with a 1:200 dilution of PE-conjugated goat anti-human IgG (Southern Biotech, #2040-09) for 1 h at room temperature. Binding was analyzed using flow cytometry by determining mean fluorescence intensities (MFIs) of PE$^{+}$-stained cell populations. FlowJo v10.4 was used for data analysis and interpretation.

**Cryo-EM sample preparation**. Prior to cryo-EM grid preparation, BG505 ΔCT T332 was concentrated to 1.6 mg ml$^{-1}$. 24 µl of the sample was mixed with 1 mM mixture of DOPC and CHS lipids (1:1, Avanti Polar Lipids). Detergent was removed by five consecutive additions of ~5 biobeads (Bio-Rad). Sample was buffer exchanged to TBS using 100 MWCO concentrators (Amicon, Millipore) prior to grid preparation and concentrated to 0.18 mg ml$^{-1}$. A thin layer of continuous carbon was deposited manually on 400 mesh size copper grids with 1.2 µm hole diameter (Quantifoil). 3 µl of sample was applied on glow discharged grid, and plunge frozen to liquid ethane using manual plunger.

Purified B41 ΔCT was concentrated to 1.13 mg ml$^{-1}$ prior to cryo-EM grid preparation. 20 µl of final sample was mixed with 1 µl of 1 mM 1:1 lipid mix and subjected to detergent removal by four consecutive biobead additions with ~5 biobeads added each time and with 1 h incubation at +4 °C between each addition. Sample was frozen on plasma cleaned, 300 mesh size UltrAuFoil grids with 1.2 µM hole diameter (Quantifoil) using Vitrobot mark IV with blot force 0, 10 s wait time and 3 s blot time.

**Cryo-EM data collection**. BG505 ΔCT T332 was imaged on an FEI Talos Arctica operating at 200 keV with a Gatan K2 direct electron detector. 2335 movie micrographs were collected over a defocus range of −1.1:−5.3 µM at 41,667X magnification (incl. post-magnification) for a final pixel size of 1.20Å2/pix. Each movie micrograph was recorded for 9.5 sec consisting of 38 frames, each exposed for 250 ms for a total electron dose of 66 e-/Å2. Data collection was controlled using Leginon automated image acquisition software while frame alignment and dose weighting were performed with MotionCor2[57].

B41 ΔCT was imaged on an FEI Titan Krios operating at 300 keV with a Gatan K2 direct electron detector. 2238 movie micrographs were collected at 48,543X magnification (incl. post-magnification) for a final pixel size of 1.03 Å2/pix. Each movie micrograph was recorded for 8.5 s consisting of 34 frames, each exposed for 250 ms for a total electron dose of 62 e-/Å2. Data collection was controlled using Leginon automated image acquisition software while frame alignment and dose weighting were performed with MotionCor2[57].

**Cryo-EM data processing and analysis**. Cryo-EM data processing was performed in a similar manner for the two samples described above. Image assessment and carbon filtering were performed with EMHP[58]. The contrast-transfer-function (CTF) was fitted to each aligned micrograph using Gctf[59] and full CTF correction

was performed during each round of processing. All single particle processing including particle picking, 2D classification, 3D classification, 3D refinement, and map sharpening were performed with Relion/2.0[60]. Briefly, one or more rounds of reference-free 2D-classification followed by subset selection were performed on 4×-binned particles followed by re-centering and re-extraction of un-binned particles. Clean particle stacks were subjected to one round of 3D auto-refinement followed by one or more rounds of 3D classification without alignment into 6 or more classes followed by subset selection. Final rounds of 3D auto-refinement were performed for each set of clean particles using a loose mask representing the shape of the complex. Final reconstructions contained 67,010 and 6574 particles for the BG505 ΔCT T332 and B41 ΔCT data sets, respectively. Resolutions were estimated as the frequency at which the gold-standard FSC drops below 0.143.

Glycan visualization and pseudo-coloring was performed as follows: BG505 ΔCT T332 and JR-FL ΔCT (EMDB id 3308) EM density maps were first low-pass filtered to 7 Å to match the resolution of B41 ΔCT, which was the lowest resolution map of the three. Glycans were located based on the corresponding Asn position in previously published JR-FL ΔCT structure (pdb id: 5FUU)[23] and the structure of the BG505 ΔCT T332 presented here. Glycans were colored on the EM map surface according to their processing status. A cutoff of >75% was set for classifying glycans as either high mannose or complex type while all other positions were colored as mixed or unprocessed if this threshold was not reached.

**Model building and refinement**. Atomic model building of BG505 ΔCT T332 was initiated by preparing homology models with SWISS-MODEL[61] using BG505 SOSIP (PDB id:5V8L)[62] and JR-FL ΔCT (PDB id:5FUU)[23] as templates for gp120 and gp41, respectively. Each individual subunit was fit into the cryo-EM density map along with two copies of PGT151 Fab taken directly from JR-FL ΔCT (PDB id:5FUU) and combined into a single model using UCSF Chimera[63]. Initial refinement was performed using RosettaRelax[64] to generate of total of 318 initial models. Each model was evaluated with EMRiner[65] and MolProbity[66] and the model with the best cumulative score was selected for further refinement. Glycans were built into the map using the carbohydrate module in COOT[67] as N-linked NAG-NAG-BMAs to all known N-linked glycosylation sites based on MS analysis and subsequently trimmed to match EM map densities. Glycans making extensive contact with PGT151 Fab (N637 and N611) were taken directly from the atomic model for JR-FL ΔCT (PDB id: 5FUU). Final refinements were performed with Phenix real-space refinement[68] without NCS constraints and with secondary structure restraints along with manual building in COOT.

## Data availability

The cryo-EM reconstructions of BG505 ΔCT T332–PGT151 and B41 ΔCT-PGT151 have been submitted to the Electron Microscopy Data Bank with accession codes EMD-9030 and EMD-9062. The atomic model for BG505 ΔCT T332–PGT151 has been submitted to the PDB with accession code 6MAR. All MS data that support the finding of this study can be downloaded from the MassIVE site (https://massive.ucsd.edu/ProteoSAFe/dataset.jsp?task=058ba71b3bda4626a783e8f1504bb55a). The authors declare that all other data supporting the findings of this study are available within the article and its Supplementary Information files, or are available from the corresponding author upon request.

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

## Acknowledgements

We thank Dr. Gabriel Ozorowski for providing the B41 SOSIP cryo-EM reconstruction. This work was supported by NIH R01AI113867 (J.C.P., J.Y., and W.R.S.) and UM1 AI 100663 (J.C.P., W.R.S., A.B.W. R.T.W. and D.R.B). The authors thank IAVI and its generous funders (www.iavi.org) (J.G.).

## Author contributions

L.C, M.P., A.B.W., D.R.B., and J.C.P. designed the research. R.A., K.R., Z.B., and J.K.D. contributed equally to this work. L.C. prepared samples for MS analyses. J.K.D., L.C., C. M.D. performed the MS analyses. L.C., S.R.P, and L.H. analyzed the MS data. M.P., L.C., R.A., D.S., R.B., and J.G. purified Env proteins and performed experiments on antigenicity. K.R., Z.B., and S.M. performed structural analyses. R.T.W., W.R.S., A.B.W., J.R. Y., D.R.B., and J.C.P. supervised the project. L.C. and J.C.P. wrote the manuscript.

## Additional information

**Competing interests:** The authors declare no competing interests.

