## [Peer Review File · Nature Communications]

Reviewers' comments:

Reviewer #1 (Remarks to the Author):

The authors report in this manuscript the glycosylation profiling of HIV envelope (Env) trimers expressed in three forms, the soluble Env trimer, the membrane-bound, and the virus-associated Env trimers. A comparison of the site-specific glycosylation patterns revealed that while the virus-associated Envs and the corresponding recombinant membrane-bound Envs produced in 293F cells exhibited high similarity in the site-specific glycan processing patterns, the soluble SOSIP trimers showed significantly different glycan processing at about half of the 26-29 glycosites. In most cases, the membrane- and virus-bound Envs demonstrated more complete processing of glycans to the complex-type, which the recombinant soluble Env trimer showed less processing and contained more high-mannose type N-glycans. This tendency was found to be consistent by analysis of the respective Env trimers from three different HIV-1 strains. These results are novel and provide a correction on some previous reports claiming that virus-associated Env was less processed and contained almost high-mannose N-glycans. Despite the complexity of glycosylation at multiple glycosites, the present study further demonstrated that the site-specific glycosylation patterns are quite conserved among the membrane-bound Envs, such as the complete switch from high-mannose-type in the soluble Env form to the complex-type at N301 in the membrane-bound form. The conservative nature of the glycosylation patterns among the soluble forms vs. the membrane-bound forms was confirmed by a systematic binding analysis with a large panel of glycan-reactive broadly neutralizing antibodies. Given the fact that the HIV Env is the sole target for broadly neutralizing antibodies (bNAbs) and a majority of bNAbs are glycan-dependent in neutralization, characterization of the site-specific glycosylation patterns and the revelation of the striking difference in the glycosylation patterns in the soluble and membrane-/viral bound Env trimers are of high significance for HIV vaccine design. This study is timely and important. The observed striking difference in the glycosylation patterns between the soluble Env trimer and membrane-bound trimer, as well as the striking similarity in site-specific glycosylation between membrane-bound and virus-associated Env trimers suggest that the membrane-bound Env trimer could be a better immunogen than the soluble Env trimer for vaccine design. The experiments were carefully designed and the data were well organized and presented. The manuscript is acceptable for publication after some minor revisions indicated below.

- 1) It is intriguing to see the remarkable difference in glycan processing between the soluble and membrane-bound Env trimers. The authors suggest that the membrane-bound Env might have better access to the membrane-bound glycosyltransferases or glycosidases. While this could be one of the factors, it is not clear if the Env trimer is membrane-associated in Golgi or if the membrane-association happens after Env processing.
- 2) A number of references missed page numbers, including Refs. 5, 11, 20, 23, 24, 30, and 46.

Reviewer #2 (Remarks to the Author):

Cao et al. used a proteomics-based mass spectrometry approach to analyze glycosylation patterns of three different HIV-1 Env preparations from three different isolates. They found that the site specific glycosylation of Envs from infectious viruses closely matched that from corresponding recombinant, membrane-bound Env trimers, but were significantly different from that of recombinant soluble, cleaved (SOSIP) Env trimers. The differences in glycosylation have a strong impact on antigenicity of these Env preparations. The authors believe that their data provide a useful guide for Env-based immunogen design to induce glycan-recognizing broadly neutralizing antibodies.

This is a much-needed study that is highly relevant to HIV vaccine development. The manuscript is very well-written and should be of general interest to the reader of this journal.

Some specific comments:

1. The authors have tried to address concerns about potential bias using PGT151 for purification. In the original paper describing this antibody (Falkowska et al., *Immunity*. 2014 May 15; 40(5): 657–668.), PGT15 neutralization does not reach 100% for several isolates analyzed, including 94UG103, IAVI C22 and 92TH021. It is unclear how much PGT151 would neutralize at a saturated concentration for the three HIV-1 strains used in this study. If not 100%, there will be fully functional Env trimers from PGT151 resistant viruses that would not bind the antibody and could not be purified, and the reported glycosylation profiles would not be complete.
2. For such this type of analysis, the quality of protein samples is undoubtedly very critical. While the gel-filtration trace shown in Fig 1C for the JRFL Δ CT trimer is impressive, the gp120:gp41 ratio from the SDS-PAGE gel looks a bit odd and is different from that from Panel E for the viral Env. Also, these protein characterization data should be shown for all protein samples from all three strains used in the study, at least in the supplementary materials.
3. The legend for Fig 1 has two (b), while (e) is missing.
4. Fig 3 is too dark to read the labels. Different colors should be used.

Reviewer #3 (Remarks to the Author):

This manuscript is an extension of a previous publication where much of the HIV envelop trimer has been characterized. What is new is here that the trimer is now expressed in other systems and the variation is observed with regard to the glycosylation. The novelty is somewhat diminished, but the results are still interesting, albeit perhaps expected. It would be expected that different expression and even changing the cell broth can change

glycosylation. However, there are still issues with the manuscript that needs clarification.

1. What does the 90 glycan refer to in the abstract. There are not 90 sites and there are no specific glycans characterized.
2. The method used to characterize the glycans have been tested, not in this manuscript but a previous one, but not very rigorously. It looks like they tried it on two proteins, again in a previously manuscript. They look for mass shifts of the peptide and use that to determine the relative abundances of the different glycotypes. I don't feel that this method has been extensively tested and validated. Some explanation or additional results should perhaps be provided.
3. Endo H is used to yield the amount of high mannose type glycans. However, Endo H works not just for high mannose, but also truncated glycans that are either hybrid or complex types. Novotny's group and others have shown this in previous publications. They also showed that it works slower for the complex and not at all for the sialylated species. Nonetheless, it does work for short complex type structures. To assume that all GlnNAc containing peptide only belong to mannose type glycans is wrong.
4. The MS response of a peptide with even a single monosaccharide can be greatly diminished compared to the lone peptide. Therefore, comparing the truncated glycopeptides to peptides not containing the monosaccharide may lead to the wrong conclusion. It has been shown that a glycopeptide can have an attenuated signal by as much as an order of magnitude.
5. To assume that aspartic acid is an indication of glycosylation after PNGase F treatment is somewhat inaccurate. There are naturally occurring aspartic acid at consensus sites without glycosylation.

Title: Differential glycan processing of HIV Envelope trimer expressed as a soluble recombinant protein or on the virus surface

Authors: Liwei Cao, and et al

Manuscript #: NCOMMS-18-12966-T

Reviewer #1 (Remarks to the Author):

The authors report in this manuscript the glycosylation profiling of HIV envelope (Env) trimers expressed in three forms, the soluble Env trimer, the membrane-bound, and the virus-associated Env trimers. A comparison of the site-specific glycosylation patterns revealed that while the virus-associated Envs and the corresponding recombinant membrane-bound Envs produced in 293F cells exhibited high similarity in the site-specific glycan processing patterns, the soluble SOSIP trimers showed significantly different glycan processing at about half of the 26-29 glycosites. In most cases, the membrane- and virus-bound Envs demonstrated more complete processing of glycans to the complex-type, which the recombinant soluble Env trimer showed less processing and contained more high-mannose type N-glycans. This tendency was found to be consistent by analysis of the respective Env trimers from three different HIV-1 strains. These results are novel and provide a correction on some previous reports claiming that virus-associated Env was less processed and contained almost high-mannose N-glycans. Despite the complexity of glycosylation at multiple glycosites, the present study further demonstrated that the site-specific glycosylation patterns are quite conserved among the membrane-bound Envs, such as the complete switch from high-mannose-type in the soluble Env form to the complex-type at N301 in the membrane-bound form. The conservative nature of the glycosylation patterns among the soluble forms vs. the membrane-bound forms was confirmed by a systematic binding analysis with a large panel of glycan-reactive broadly neutralizing antibodies. Given the fact that the HIV Env is the sole target for broadly neutralizing antibodies (bNAbs) and a majority of bNAbs are glycan-dependent in neutralization, characterization of the site-specific glycosylation patterns and the revelation of

the striking difference in the glycosylation patterns in the soluble and membrane-/viral bound Env trimers are of high significance for HIV vaccine design. This study is timely and important. The observed striking difference in the glycosylation patterns between the soluble Env trimer and membrane-bound trimer, as well as the striking similarity in site-specific glycosylation between membrane-bound and virus-associated Env trimers suggest that the membrane-bound Env trimer could be a better immunogen than the soluble Env trimer for vaccine design. The experiments were carefully designed and the data were well organized and presented. The manuscript is acceptable for publication after some minor revisions indicated below.

We thank the reviewer for these positive comments.

1) It is intriguing to see the remarkable difference in glycan processing between the soluble and membrane-bound Env trimers. The authors suggest that the membrane-bound Env might have better access to the membrane-bound glycosyltransferases or glycosidases. While this could be one of the factors, it is not clear if the Env trimer is membrane-associated in Golgi or if the membrane-association happens after Env processing.

For clarification about the association of Env with the ER/Golgi membrane we have added the following text and references 43-45 to the discussion (Line 327-334):

“During their synthesis the soluble SOSIP trimer is released into the lumen of the ER while the membrane bound (Δ CT) Env remains bound to the membrane. Thus, their topology are different during their transit through the ER and Golgi apparatus where the glycan-processing enzymes and glycosyltransferases are membrane-bound.”

Moreover, we did not intend to suggest that the glycans on the membrane bound forms have better access to the processing enzymes, only that the access/topology of the soluble and membrane bound forms are different. We made small modifications to the discussion and legend of Fig. 6 to further clarify.

2) A number of references missed page numbers, including Refs. 5, 11, 20, 23, 24, 30, and 46.

Thank you for catching these reference errors, which we have corrected in the revised manuscript.

Reviewer #2 (Remarks to the Author):

Cao et al. used a proteomics-based mass spectrometry approach to analyze glycosylation patterns of three different HIV-1 Env preparations from three different isolates. They found that the site specific glycosylation of Envs from infectious viruses closely matched that from corresponding recombinant, membrane-bound Env trimers, but were significantly different from that of recombinant soluble, cleaved (SOSIP) Env trimers. The differences in glycosylation have a strong impact on antigenicity of these Env preparations. The authors believe that their data provide a useful guide for Env-based immunogen design to induce glycan-recognizing broadly neutralizing antibodies.

This is a much-needed study that is highly relevant to HIV vaccine development. The manuscript is very well-written and should be of general interest to the reader of this journal.

We thank the referee for these positive comments.

Some specific comments:

1. The authors have tried to address concerns about potential bias using PGT151 for purification. In the original paper describing this antibody (Falkowska et al., Immunity. 2014 May 15; 40(5): 657–668.), PGT15 neutralization does not reach 100% for several isolates analyzed, including 94UG103, IAVI C22 and 92TH021. It is unclear how much PGT151 would neutralize at a saturated concentration for the three HIV-1 strains used in this study. If not 100%, there will be fully functional Env trimers from PGT151 resistant viruses that would not bind the antibody and could not be purified, and the reported glycosylation profiles would not be complete.

The referee raises a valid concern, namely, if PGT-151 did not bind to fully functional Env variants, we could miss variant glycoforms that were not recognized by the antibody and introduce bias into our results. Directly to the point of the reviewer, we also had this concern and had measured neutralization of PGT151 to all three viruses, JR-FL E168K, BG505, and B41. We found that JR-FL and BG505 viruses were 100% neutralized by PGT151, while B41 virus was neutralized to a plateau of ~75%. The results indicate that there is no concern for bias using PGT-151 for purification of Env from JR-FL and BG505, and the fact that results for B41 were complementary, there is also little concern for significant bias even though the neutralization was only 75%. We have now added the neutralization results to supplementary information as Figure S5, and mention them in discussion of the justification of using PGT151 for purification. (see Line 370-373)

2. For such this type of analysis, the quality of protein samples is undoubtedly very critical. While the gel-filtration trace shown in Fig 1C for the JRFL Δ CT trimer is impressive, the gp120:gp41 ratio from the SDS-PAGE gel looks a bit odd and is different from that from Panel E for the viral Env. Also, these protein characterization data should be shown for all protein samples from all three strains used in the study, at least in the supplementary materials.

We very much agree with the reviewer that the quality of the protein samples is key. With regards to the concern about the apparent gp120:gp41 ratio, in the non-reduced SDS-PAGE gel in Figure 1D, in the original gel the bands for gp41 and PGT151 Fab were close to each other, and the Fab fragment was dominant making the ratio appear 'odd'. To address this concern we have replaced the original panel with a new panel, including both reducing and nonreducing SDS-PAGE of JR-FL ΔCT, in which the gp41 band and the fab band are distinguished from each other.

In addition, we have now added the protein characterization data for all proteins analyzed in this study to the supplementary materials as Figure S6 .

3. The legend for Fig 1 has two (b), while (e) is missing.

Thank you, we have corrected the legend of Figure 1.

4. Fig 3 is too dark to read the labels. Different colors should be used.

We agree that the labels need to be clearly legible. In our previous publication we used the same colors, and the labels were actually smaller font, yet in the published version, the labels are very clear (Cao et al. (2017) Nat. Comm. 8, 14954, PMID: 28348411).

In the pdf of this manuscript the labels are somewhat difficult to read, in part due to the conversion to reduce the file size. However, the original looks very legible, even more so than the original artwork for our previous manuscript. We will work with the journal to ensure that the labels in the final figure are easy to read.

Reviewer #3 (Remarks to the Author):

This manuscript is an extension of a previous publication where much of the HIV envelop trimer has been characterized. What is new is here that the trimer is now expressed in other systems and the variation is observed with regard to the glycosylation. The novelty is somewhat diminished, but the results are still interesting, albeit perhaps expected. It would be expected that different expression and even changing the cell broth can change glycosylation.

We are pleased that reviewer finds our work interesting despite concerns about novelty. A major aspect of the novelty is that we assess for the first time the site specific glycosylation of the HIV Env trimer from three different virus strains expressed in the natural host cells, peripheral blood mononuclear cells (PBMCs). The novelty and interest in this achievement stems from this biological context, providing critical information for vaccine design, where the goal is to develop a vaccine that is most “native like” to achieve the goal of inducing broadly neutralizing antibodies to HIV.

We agree with the referee that differences in glycosylation might be expected when comparing glycoproteins produced in different cells. But surprisingly, what we find is the opposite, namely, that the site-specific glycan processing of Env purified from virus produced in PBMCs is highly similar to pseudovirus and recombinant membrane bound Env (Δ CT) produced in HEK293 cells. In contrast, the site-specific glycan processing of the soluble SOSIP trimers in HEK293 cells is quite different at 50% of the sites. Remarkably, these observations are the same for the Envs of all three viruses examined. Thus, we believe that these unexpected findings are important for the rational design of HIV vaccines.

However, there are still issues with the manuscript that needs clarification.

1. What does the 90 glycan refer to in the abstract. There are not 90 sites and there are no specific glycans characterized.

HIV envelope glycoprotein (Env) is a trimer of comprised of monomers that are cleaved into gp120 and gp41 subunits associated through non-covalent interaction. Depending on the strain, monomer contains 25-30 N-linked glycosylation sites. Thus Env trimer contains up to 90 N-linked glycosylation sites (N-glycans). To clarify what we meant by “90 N-glycans” we have reworded the sentence in the abstract/summary as follows: “Because HIV Env is densely glycosylated with 75-90 N-glycans per trimer, most bnAbs use or accommodate them in their binding epitope, making the glycosylation of recombinant Env a key aspect of HIV vaccine design.”

2. The method used to characterize the glycans have been tested, not in this manuscript but a previous one, but not very rigorously. It looks like they tried it on two proteins, again in a previously manuscript. They look for mass shifts of the peptide and use that to determine the relative abundances of the different glycotypes. I don't feel that this method has been extensively tested and validated. Some explanation or additional results should perhaps be provided.

We now have two published manuscripts on the method. The first is the description of the method as mentioned by the reviewer (Cao et al. (2017) Nat. Comm. 8, 14954, PMID: 28348411), and the second is a detailed protocol that has been recently published (Cao et al. (2018) Nat. Protocol. 13, 1196-1212, PMID: 29725121). Between the two manuscripts, the method has been extensively validated, and has been tested on ten different proteins, including yeast invertase, serum glycoproteins (α_1 -acid glycoprotein, transferrin, fetuin), antibodies (IgG1, IgG2a, IgM) and virus glycoproteins (MERS coronavirus, influenza virus hemagglutinin, HIV Env).

The method relies on sequential digestion with Endo H to quantitatively remove high mannose/hybrid glycans, followed by PNGase F to remove complex type glycans. For yeast invertase, known to contain only high mannose type glycans, the method shows only oligomannose glycans, demonstrating that Endo H quantitatively removes oligomannose glycans, producing GlcNAc-peptide (N+203) prior to the treatment with PNGase F. For glycoproteins known to contain only complex type glycans (e.g. α_1 -acid glycoprotein, transferrin, fetuin), the method shows only complex type glycans, showing that prior treatment with Endo H does not remove any complex type glycans, and generating peptide with Asp (N+3). In both publications we report a similar validation experiment using the soluble HIV SOSIP Env produced in the presence of kifunensine that prevents processing to complex type glycans. Treatment with Endo H, then PNGase F shows only high mannose type glycans (N+203). However, if Endo H treatment is omitted, subsequent treatment with PNGase F results in all glycans being removed, and the result is as if all peptides had complex type glycans (Asn->Asp, N+3). This validation step also allowed us to compare the sensitivity for detection of the alternative GlcNAc or Asp modified peptides as discussed further below for concern #4.

3. Endo H is used to yield the amount of high mannose type glycans. However, Endo H works not just for high mannose, but also truncated glycans that are either hybrid or complex types. Novotny's group and others have shown this in previous publications. They also showed that it works slower for the complex and not at all for the sialylated species. Nonetheless, it does work for short complex type structures. To assume that all GlcNAc containing peptide only belong to mannose type glycans is wrong.

On this point we respectfully disagree with the referee. Indeed, it is well documented in the literature that Endo H has a strict specificity for cleavage of high mannose/hybrid structures, and not complex type glycans. In fact many other investigators routinely use Endo H to assess the oligomannose/hybrid content of glycoproteins (e.g. see Hudson H. Freeze, Curr Protoc Mol Biol, 2010; & Laura K. Pritchard, Nature Communications, 2015 as examples). We have now clearly stated this assumption at the beginning of the results section.

Furthermore, as cited above in point 2, we tested this key assumption in our validation experiments. Endo H removed no glycans from glycoproteins that contain only complex glycans, not even small, truncated complex type glycans such as those found in antibodies. For all such proteins, no GlcNAc-peptides (N+203) were identified.

To further address the concern of the reviewer about alternative specificities of Endo H, we conducted a thorough literature search for examples of side activities of Endo H and found none. Since the work of Novotny was cited as an example, we did a PubMed search for Novotny AND Endo, and separately Novotny AND endoglycosidase. Both searches gave a single relevant hit, for a manuscript entitled “Assigning N-glycosylation sites of glycoproteins using LC/MSMS in conjunction with endo-M/exoglycosidase mixture” (Segu et al. (2010) J. Proteome Res. 9, 3598). This 'Novotny' report studies Endo M, which is a totally different endoglycosidase, not related to Endo H. Endo M does indeed have the specificity for removing truncated and complex glycans as mentioned by the reviewer. However, this is not at all relevant to the specificity of Endo H, or to the use of Endo H by us and others as a tool to assess the oligomannose content of glycoproteins. Perhaps this work of Novotny with Endo M was the basis of the concern of the reviewer about the specificity of Endo H.

4. The MS response of a peptide with even a single monosaccharide can be greatly diminished compared to the lone peptide. Therefore, comparing the truncated glycopeptides to peptides not containing the monosaccharide may lead to the wrong conclusion. It has been shown that a glycopeptide can have an attenuated signal by as much as an order of magnitude.

The reviewer expresses a concern that was at one time a major concern of the field, namely that glycan-modified peptides would be detected with low efficiency. However, this is no longer a generally held view. In fact, the MS/MS glyco-proteomics approaches used by the Crispin and Desaire groups for site specific analysis of glycosylation for HIV Env search for intact glycopeptides that contain N-linked glycans from 7-17 or more monosaccharide units, with the assumption that spectral hits are detected with similar efficiency regardless of glycan size.

Our method was based on a previous report by the Kolarich group looking at the detection efficiency of synthetic peptides containing Asn (N+0), Asp (N+3) or GlcNAc-Asn (N+203), and demonstrating that they were recognized with similar efficiency during ESI-MS analysis. (Stavenhagen, K. (2013) J. Mass Spectrom. 48, 627). This work demonstrated that mass spectrometry could be used as a basis for the semi-quantitative approach we developed using endoglycosidases to generate peptides with glycosites containin either Asn (N+0; no glycan), Asp (N+3; complex type glycan) or GlcNAc-Asn (N+203; high mannose/hybrid glycan). To better orient the reader to this approach we have added an additional explanation of our method to the results section (Line 111-125).

During validation of our approach we directly compared the ionization efficiency of peptides with N+3 and N+203 modifications. This validation step is already mentioned in item 2 where HIV Env was produced in the presence of kifunensine (Kif) to produce high mannose glycans. Use of Env produced in the presence of Kif allowed us to generate either GlcNAc-Asn (N+203) at all 28 glycosites using Endo H, or Asp (N+3) at all 28 glycosites using PNGase F. Then by mixing the peptide products of both digests in a 1:1 ratio prior to MS/MS analysis, we could directly assess the efficiency for detection of the two species at each of the 28 glycosites (Cao L.W., Nature Communications, 2017 & Cao L.W., Nature Protocols, 2018). We

showed that the two versions (N+3 and N+203) of same peptide (except at glycosylation site) display similar ionization efficiency during mass spectrometry analysis, and that applying a site specific 'correction factor' minimally changed the result at any glycosite compared to no correction. Thus, we feel we have directly addressed this concern during validation of our method.

5. To assume that aspartic acid is an indication of glycosylation after PNGase F treatment is somewhat inaccurate. There are naturally occurring aspartic acid at consensus sites without glycosylation.

It is certainly the case that not all DNA encoded glycosites (Asn-X-Thr/Ser) are glycosylated. In fact, perhaps only 40-50%. It is also the case that Asn can undergo spontaneous oxidation to Asp (N+1) before or during work up of peptide digests. However, this is not a concern in our analysis for several reasons: 1) As we show in this report, most of the 26-28 glycosites (per monomer) are fully glycosylated, preventing spontaneous oxidation to aspartic acid (Asp). So except for a couple of sites, there is not a significant amount of Asn not already occupied by glycan. 2) Spontaneous conversion of Asn to Asp would appear as N+1. We perform the PNGase F digestion in O¹⁸ water, which converts Asn to Asp with N+3, which would not be confused with spontaneous conversion to Asp with N+1. Early in the development of the method we were concerned about this and assessed the amount of Asn to Asp (N+1) and found negligible amounts. Taking these points into account, we believe that this is not a significant concern in our results.

REVIEWERS' COMMENTS:

Reviewer #3 (Remarks to the Author):

The authors have addressed well the concerns of this reviewer. The manuscript should be accepted for publication.

REVIEWERS' COMMENTS:

Reviewer #3 (Remarks to the Author):

The authors have addressed well the concerns of this reviewer. The manuscript should be accepted for publication.

A: We thank the reviewer for these positive comments.